# Application of Flow Cytometry in the Diagnosis of Bovine Epidemic Disease

**DOI:** 10.3390/v15061378

**Published:** 2023-06-15

**Authors:** Zhilin Liu, Yuliang Zhang, Donghui Zhao, Yunjiao Chen, Qinglei Meng, Xin Zhang, Zelin Jia, Jiayu Cui, Xueli Wang

**Affiliations:** 1College of Animal Science and Technology, Inner Mongolia Minzu University, Tongliao 028000, China; liuzhilin1998@163.com (Z.L.); chenyj032899l@163.com (Y.C.); mry042815@163.com (Q.M.); zx15144845347@163.com (X.Z.); zl15857870921@163.com (Z.J.); 15750457174@163.com (J.C.); 2Tongliao City Animal Quarantine Technical Service Centre, Tongliao 028000, China; 13804753385@163.com (Y.Z.); 18747530057@163.com (D.Z.)

**Keywords:** flow cytometry, bovine epidemic diseases, diagnose

## Abstract

As science and technology continue to advance, the use of flow cytometry is becoming more widespread. It can provide important information about cells in the body by detecting and analysing them, thereby providing a reliable basis for disease diagnosis. In the diagnosis of bovine epidemic diseases, flow cytometry can be used to detect bovine viral diarrhoea, bovine leukaemia, bovine brucellosis, bovine tuberculosis, and other diseases. This paper describes the structure of a flow cytometer (liquid flow system, optical detection system, data storage and analysis system) and its working principles for rapid quantitative analysis and sorting of single cells or biological particles. Additionally, the research progress of flow cytometry in the diagnosis of bovine epidemic diseases was reviewed in order to provide a reference for future research and application of flow cytometry in the diagnosis of bovine epidemic diseases.

## 1. Introduction

With the improvement of people’s living standards, the scale of cattle breeding has been expanding, and the number of cattle being bred has increased sharply. Some problems that are not conducive to the development of cattle breeding have been exposed, especially the occurrence of bovine epidemic diseases (including brucellosis, bovine viral diarrhoea, bovine leukaemia, bovine tuberculosis, etc.), which have brought huge losses to the cattle breeding industry. Therefore, knowing how to quickly and accurately diagnose cattle epidemics is particularly important.

Currently, diagnostic techniques for bovine epidemic diseases include enzyme-linked immunosorbent assay (ELISA), serum (virus) neutralisation test, virus isolation, fluorescent antibody staining technique, PCR method, fluorescent PCR, loop-mediated isothermal amplification (LAMP), etc. These techniques have advantages and disadvantages in the detection of bovine epidemic diseases [1,2]. For example, the traditional PCR method is fast and simple, but the operation is prone to systematic errors and can only achieve semi-quantitative results. Although the ELISA has a high degree of specificity and sensitivity, it is susceptible to a large number of false-positive results due to many influencing factors [3].

Flow cytometry is a technique for rapid multi-parameter analysis of single cells in solution that can be used in immunology, molecular biology, bacteriology, virology, cancer biology, and infectious disease surveillance [4]. Due to the advantages of high resolution, a large number of resolved cells, many parameters, high accuracy, and high speed, flow cytometry has been widely used in the fields of disease diagnosis, tumour detection, and immunoassay in recent years, especially as research on the detection of bovine epidemic diseases has developed rapidly [5]. In this paper, we have reviewed the progress of flow cytometry in the diagnosis of bovine epidemic diseases based on relevant studies at home and abroad in order to provide a reference for the further development and application of this technology in the diagnosis of bovine epidemic diseases.

## 2. Flow Cytometry (FCM)

### 2.1. Definition and Characteristics of Flow Cytometry

FCM is a technique that uses a flow cytometer to simultaneously perform multi-parameter, rapid quantitative analysis and sorting of cells or biological particles in a fast linear flow state [6]. The characteristics of flow cytometry are as follows: (1) The detection speed is fast, as tens of thousands of single biological particles or other particles in the solution can be detected in a short time, and a large number of particles can be evaluated in a very short time, so the results are statistically strong. (2) There are numerous detection parameters, and flow cytometry can simultaneously detect the scattered light and various fluorescence signals of a single biological particle. Although the particle detection speed can reach 100,000 times per second, each particle can also collect and analyse 20 parameters simultaneously. (3) Rapid analysis of single biological particles and accurate sorting occur at the same time. Sorting is a process of cell particle classification that can physically separate single particles or cells from mixed populations at extremely fast detection speeds and retain the viability of most cells [7]. In addition, in vivo flow cytometry has been developed. In vivo flow cytometry (IVFC) is a promising new tool for counting circulating cells that can continuously measure circulating cell populations over a period of time and, at the same time, detect and quantify circulating fluorescently labelled cells in living animals in real time. The advantages of this method are that there is no need to draw blood samples, it has high sensitivity, and it can detect cells at very low concentrations [8,9,10,11].

### 2.2. Structure and Principles of Flow Cytometry

The basic structure of flow cytometry mainly includes a liquid flow system, an optical detection system, data storage, and an analysis system (as shown in Figure 1). In addition, some flow cytometers with sorting functions also have sorting systems [12]. The principle of flow cytometry is based on the scattering of light and the emission of fluorescence. When individual cells in a hydrodynamically focused cell suspension interact with a laser beam, the light scatter and fluorescence signals emitted by the cells are collected by a detector, which converts these signals into electrical currents. The electronics convert these analogue current signals into digital signals and finally visualise the data in the form of histograms or dot plots [13,14].

## 3. Application of Flow Cytometry in the Diagnosis of Bovine Epidemic Diseases

Flow cytometry is a cutting-edge technology that integrates cell staining, sorting, and flow cytometric detection to accurately and rapidly classify a variety of cell types. By screening, counting, and analysing different cells, it can determine the cause and mode of transmission of diseases and provide an important reference for disease diagnosis, prevention, and control. In recent years, flow cytometry has been widely used in the detection of bovine epidemic diseases (as shown in Table 1) and has played an important role in the diagnosis of bovine viral diarrheal mucosal disease (BVDV), bovine endemic leukaemia (EBL), bovine brucellosis, bovine tuberculosis (BTB), and other epidemic diseases.

### 3.1. Application in the Diagnosis of Bovine Viral Diarrhoea

Bovine viral diarrhoea is a disease caused by the bovine viral diarrhoea virus (BVDV) and is characterised by fever, diarrhoea, respiratory symptoms, and reproductive disorders, which can cause peripheral blood lymphopenia and lymphocyte apoptosis [27,28]. BVDV belongs to the plague virus genus of the Flaviviridae family. The viral genome is a single-stranded RNA, encoding four structural proteins (i.e., C, Erns, E1, and E2) and eight non-structural proteins (i.e., Npro, p7, NS2, NS3, NS4A, NS4B, NS5A, and NS5B) [29,30]. Dong Kun et al. [31] applied a double antibody sandwich ELISA kit for bovine viral diarrhoea virus and bovine enterovirus antigen to 738 faecal samples collected from five regions in Jilin Province in 2019–2020, respectively, and confirmed some samples with PCR and immunofluorescence tests. The results showed that the prevalence of BVDV infection in cattle of different regions, ages, and breeds in Jilin Province was 21.68%. Chang Liyun et al. [32] collected 788 faecal samples of calves with diarrhoea from 38 dairy farms in different areas of Tangshan City and used PCR to detect the E2 gene of the bovine viral diarrhoea virus in faecal samples. The results showed that the positive detection rate for BVDV was 29.82% (235/788). According to the above data, the incidence of BVD in China has been relatively high in recent years and has affected the healthy development of the cattle breeding industry in China. Therefore, a timely and accurate diagnosis of the disease is extremely important. Currently, real-time PCR and enzyme-linked immunosorbent assays (ELISAs) can effectively identify BVDV, but it is difficult to detect and quantify viral infection at the single cell level. BVDV infection is accompanied by immune regulation, and it is crucial to detect it at the level of individual lymphocytes.

Falkenberg et al. [15] investigated a novel PrimeFlow RNA assay for in situ detection of BVDV. The model included three BL-3 cell lines with different infection states: one without BVDV infection (NADC-BL3-SF), one with BVDV infection (CRL-2306), and one with both BVDV and bovine leukaemia virus (CRL8037). BVDV RNA in BVDB-infected cell lines was detected by flow cytometry using an RNA probe specific for the BVDV-2a Npro-Erns coding region. The results showed that no BVDV-RNA-positive cell population was detected in the NADC-BL3-SF cell line, but 60% of the cells in the CRL8037 cell line and 90% of the cells in the CRL-2306 cell line were BVDV-RNA-positive. This is the first report of in situ detection of BVDV at the single-cell level.

Falkenberg et al. [16] evaluated the cell-mediated response in inoculated calves using the PrimeFlow RNA assay, which combines cell surface marker staining and intracellular cytokine RNA expression. The results showed that in both CD2^+^ and CD4^+^ T cell subsets, the expression of CD25 was significantly increased by 5.35 in vaccinated calves compared with unvaccinated calves. At the same time, the expression of IFN-γ in CD2^+^, CD4^+^, CD8^+^, and CD335^+^ T cell subsets in the vaccinated group was significantly higher than that in the unvaccinated group, which increased by 2.44, 3.21, 2.98, and 15.85, respectively. This is consistent with previous findings that cattle vaccinated with MLV vaccine containing BVDV exhibited a recall memory response after BVDV restimulation by inducing upregulation of CD25 (IL-2 receptor) and an increase in intracellular IFN-γ protein in T cell subsets [33]. 

Persistent infection with BVDV is the result of immune tolerance and is not usually associated with lymphopenia, but acute BVDV infection results in systemic immunosuppression characterised by depletion of circulating lymphocytes and CD4^+^ and CD8^+^ T cells. Grandoni et al. [17] infected 203-day-old artificially inseminated buffalo with BVDV-1, and flow cytometry detection was performed at 0, 3, 4, and 14 days after infection (dpi). The results showed that, compared to the control group, the infection group had a general decrease in all lymphocyte subpopulations, leading to a severe reduction in lymphocytes. T lymphocytes (CD3^+^) decreased significantly at 3 and 4 dpi, with a maximum decrease of 55.8%. CD4^+^ and CD8^+^ T cells decreased by 53% and 56.7%, respectively, reaching their lowest levels at 4 dpi. The greatest average decrease in NK cells was observed at 4 dpi (−71.5%). Therefore, acute BVDV can be initially diagnosed by a significant decrease in the number of lymphocytes.

Detection of BVDV at the single lymphocyte level is important for studying peripheral blood mononuclear cell subsets during BVDV infection, but there are few methods to detect and quantify BVDV at this level. PrimeFlow RNA is a novel assay for the detection of BVDV. The use of this assay to further evaluate the viral RNA in the original specimens of BVDV-infected animals and the properties of different cell subsets that may be affected during infection can help to study the immune response associated with BVDV. Although the assay has a certain sensitivity, further studies are needed on samples collected from live animals. In addition, the detection of lymphocytes by flow cytometry can be used to preliminarily identify acute BVDV.

### 3.2. Application in the Diagnosis of Bovine Endemic Leukaemia

Bovine leukaemia virus (BLV) is the causative agent of endemic bovine leukaemia and belongs to the genus Deltavirus, family Retroviridae. The virus infects lymphocytes and integrates DNA intermediates into the cellular genome as a provirus. The disease is a chronic neoplastic disease of cattle, and most infected cattle are asymptomatic, which makes the virus have a high shedding rate in many cattle herds, posing a serious threat to the development of cattle farming [34,35,36]. The common detection methods for BLV include qPCR, ELISA, and lymphocyte count. With the development of flow cytometry, flow technology has also been applied to the detection of this pathogen in recent years [37].

Telomere shortening and increased telomerase activity are two of the characteristics of many haematological neoplastic diseases. Szczotka et al. [18] washed the mixture of lymphocytes (1 × 10^6^ cells/mL) and the 1301 cell line (1 × 10^6^ cells/mL) and suspended them together in 300 µL of hybridization buffer with a PNA/FITC telomere probe. Hybridization was performed after denaturation at 82 °C for 10 min and overnight at room temperature in the dark. After hybridization, the cells were washed, resuspended in propidium iodide and an RNase A staining solution, and incubated at 4 °C for 3 h before detection by flow cytometry. The research showed that the telomere fluorescence intensity of BLV-infected cattle was significantly lower than that of healthy cattle, and the relative telomere length (RTL) value of BLV-positive bovine cells (31.63 ± 12.62) was lower than that of the control group (38.43 ± 4.03). The above studies indicate that flow cytometry can be used to detect BLV infection in cattle by measuring telomere length.

It has been reported that in BLV cattle, a large number of cells can express both IgM and BLV-structural glycoprotein 51 (BLV-gp51) antigens. This study was confirmed by Szczotka et al. [19], who used flow cytometry to detect the expression of BLV gp51 in the blood of naturally infected cows. Blood samples were taken from positive cows and cultured for a short period. Cells were counted and sorted by flow cytometry (bovine lymphocyte levels between 7600 and 11,200 cells/mm^3^ were classified as AL; bovine lymphocyte levels between 12,800 and 63,200 cells/mm^3^ were classified as PL), and flow cytometry was used to analyse lymphocyte subsets. Findings showed that BLV-gp51 was mainly expressed in CD19^+^ and CD19^+^ IgM^+^ lymphocytes. The proportion of CD19^+^ B cells expressing BLV-gp51+ antigen increased with the progression of BLV infection, from 21.0% in the AL phase to 28.25% in the PL phase, and the proportion of CD19^+^ IgM^+^ B cells expressing BLV-gp51+ antigen increased from 11.4% in the AL group to 17.7% in the PL group. This experiment showed an increase in gp51 expression with disease duration. Therefore, BLV infection could be confirmed by detecting the expression of BLV gp51 in cattle by flow cytometry. 

Iwan et al. [20] evaluated the effect of BLV infection on DC characteristics by taking bovine peripheral blood to isolate individual nucleated cells and isolating bone marrow dendritic cells (MDCs) from the individual nucleated cells, which were finally analysed using flow cytometry. The results showed that the level of IL-10 was significantly increased compared to the negative control group. The average concentration of IL-10 in the positive group was 2.98 pg/mL, and that in the control group was 1.48 pg/mL. IFN-γ levels were significantly reduced in BLV-infected MDCs, with mean concentrations of 1.16 pg/mL in the infected group and 2.35 pg/mL in the control group. IL-10 can actively block BLV replication in adherent monocyte/macrophage cultures, suggesting a role for this cytokine in the viral latency mechanism [38]. Other studies have reported similar inverse correlations between IL-10 and IFN-γ in sera from BLV-infected cattle. Therefore, the contents of IL-10 and IFN-γ in MDCs can be detected to further determine whether they are infected with BLV. 

A telomere is a nuclear protein structure found at the end of each chromosome arm that is composed of highly conserved hexameric (TTAGGG) tandem repeat DNA sequences. Its function is to maintain the stability of the genome, and the stability of telomere length is essential for normal cell function [39,40]. However, telomerase is a DNA polymerase that lengthens the 3′ end of the chromosome by stepwise synthesis of multiple telomeric repeats, which is activated and involved in telomere maintenance in many cancer cells [41,42]. Telomere shortening and increased telomerase activity can be markers of BLV infection, as observed by Hemmatzadeh et al. [43]. The detection of telomere length and telomerase activity, gp51 expression, and IL-10 and IFN-γ production in dendritic cells by flow cytometry can further improve the sensitivity of BLV detection and, hence, the timely treatment of affected cattle and reduce the economic losses caused by BLV infection.

### 3.3. Application in the Diagnosis of Bovine Brucellosis

In recent years, the incidence of brucellosis in cattle has gradually increased. Brucellosis is an important zoonotic infectious disease that can cause reproductive disorders in cattle and has a huge economic and public health impact on the world [44,45]. The common serological diagnosis of bovine brucellosis can be tested in two ways: the rose Bengal plate agglutination test and the complement fixation test. These two tests mainly detect S-LPS, an antibody against smooth lipopolysaccharide, but both methods cannot avoid false-positive serological reactions (FPSRs) caused by the detection of bacteria with common S-LPS components of Brucella and lead to the single reactor (SR) phenomenon [46]. Therefore, there is an urgent need for a method to diagnose bovine brucellosis that can avoid false-positive serologic reactions. 

Grandoni et al. [21] distinguished between positive and negative buffalo by assessing lymphocyte subpopulation variation in buffalo naturally infected with brucellosis. A 50 μL whole blood sample was incubated with saturated concentrations of anti-CD18, CD21, and CD335 monoclonal antibodies at 4 °C in the dark for 20 min. After washing and centrifugation, cells were fixed and infiltrated, resuspended, cultured with anti-CD3e and CD79a monoclonal antibodies, and centrifuged again. The samples were resuspended in 120 μL of cold PBS for flow cytometry analysis. The results showed a significant reduction in total lymphocytes, B-lymphocytes, and T-lymphocytes in the positive animals. PBMC in the positive group decreased from 40.3 ± 1.1 to 29.5 ± 1.1; total lymphocytes, T lymphocytes, and B lymphocytes decreased from 35.5 ± 1.1 to 23.0 ± 1.2, 28.9 ± 1.0 to 19.2 ± 1.1, and 5.7 ± 0.3 to 2.6 ± 0.3, respectively. This was consistent with the results of E. Crosby et al. [47] in the study of blood from patients with brucellosis.

Agnone et al. [22] diagnosed the disease by observing the expansion of interferon-γ (IFN-γ^+^) T cell subsets in peripheral blood mononuclear cells (PBMC) from cattle infected with Brucella. PBMC were isolated from heparinized blood samples. Samples containing 4 × 10^5^ cells/mL PBMs were placed in a 5% CO_2_ environment and cultured at 37 °C for 48 h. PBMC were collected and stained with fluorescein isothiocyanate (FITC)-labelled mAbs (anti-CD8, anti-CD4, and -WC1). After incubation at 4 °C for 30 min, PBMC were fixed, permeated, added with PE-labelled anti-IFN-γ mAb, cultured at 4 °C for 30 min, washed, resuspended, and finally detected using flow cytometry. The results showed that compared with the control group, the percentage of IFN-γ^+^ T lymphocytes increased in the positive group, and the percentage of IFN-γ^+^ CD4^+^ T lymphocytes increased significantly. CD4^+^ IFN-γ^+^ T cell subsets are thought to be required for clearance of brucella infection and activation of macrophage killing mechanisms in mouse models [48].

Boggiatto et al. [23] have developed a method based on flow cytometry to gain a better understanding of the response of Brucella-specific T cells. Peripheral blood samples were collected from cattle vaccinated with the Brucella abortus RB51 strain vaccine. PBMC were isolated and labelled. The RB51 antigen was used in a two-step in vitro stimulation test and finally stained for cell surface receptors and intracellular cytokines. A statistically significant difference in the frequency of CD4^+^ T cell proliferation and IFN-γ production in response to RB51 antigen stimulation was observed between control and vaccinated animals. About 21% of the CD4^+^ T cells in the control group proliferated or produced IFN-γ in vitro, whereas 32% of the CD4^+^ T cells in the vaccination group responded to antigen stimulation, 45.5% of them proliferated and produced IFN-γ, and 35.6% produced only IFN-γ. This is consistent with the results of previous studies, and therefore bovine brucellosis can be diagnosed by this method.

At present, several flow cytometry methods based on intracellular CD4^+^ IFN-γ^+^ T assays have been developed to identify bovine brucellosis and avoid false-positive serum reactions. However, only a few experiments have used this method in cattle to describe Brucella-specific responses, so continuous optimisation and a large number of animal tests are needed to demonstrate its effectiveness.

### 3.4. Application in the Diagnosis of Bovine Tuberculosis

Bovine tuberculosis (TB) is a chronic bacterial disease caused by Mycobacterium bovis and a major concern for public health. The disease can infect not only cattle but also badgers, deer, goats, pigs, camelidae (alpacas), dogs, humans, and other mammals [49,50]. Xu Fang [51] tested a total of 6943 cattle from 185 farm herds in 14 provinces as well as the Xinjiang Production and Construction Corps for bTB SICCT from 2015–2018, respectively. The results showed that the individual positivity rates for bTB SICCT were 17.43%, 25.09%, 28.02%, and 21.62% from 2015–2018, respectively. Wang Bingqing [52] conducted a bovine tuberculosis survey on seven large-scale dairy farms in Suining County, Xuzhou City, Jiangsu Province, during the period 2018–2020. The survey results show that the bTB positivity rate was 2.28%, 12.21%, and 8.83% in 2018, 2019, and 2020, respectively. From the above data, it can be seen that although the prevalence of bovine tuberculosis varies across China, the situation is still serious and not optimistic. The most effective way to control the disease is to kill infected animals. Therefore, detection technology has become an important factor in the control of bovine tuberculosis [53]. Currently, the main methods used to detect BTB are the IFN-γ in vitro release test, the tuberculin intradermal hypersensitivity test (TST), and the antibody detection method. However, all three methods have some limitations, as the TST has low sensitivity and specificity, is prone to false positives and false negatives, and cannot distinguish between latent infection and active tuberculosis. Although the antibody detection method has high specificity, is rapid, simple, and low-cost, and tends to detect animals in the late stage of infection, it has low sensitivity and may miss animals in the early stage of infection [54]. When the in vitro release test of IFN-γ was used as an auxiliary detection method for bovine tuberculosis, the search for new biomarkers and new detection methods for bovine tuberculosis became a new direction for the diagnosis of bovine tuberculosis [55]. Due to its excellent characteristics of high sensitivity, high specificity, high precision, and high yield, flow cytometry is considered a new method for the detection of bovine tuberculosis [56].

Xia A et al. [24] isolated bovine peripheral blood mononuclear cells (PBMs) and seeded 4 × 10^5^ cells/mL of PBMs in RPMI medium in 24-well plates. At the same time, avian tuberculin (PPDA), bovine tuberculin (PPDB), and CFP-10-ESAT-6 fusion protein (CE) were added to 24-well plates and incubated at 37 °C in 5% CO_2_ for 24 h. Brefeldin A was added after 4 h of incubation. PBMs were then washed and surface stained with the corresponding CD4 antibody, followed by fixation and infiltration, and finally indirect staining with monoclonal antibodies against BoIL-2 and Alexa Fluor 488-labelled goat anti-mouse IgG H&L (Figure 2). The results showed that the CD4^+^ T cells of BTB-infected animals with endocrine antigen-specific IL-2 were significantly higher than those of healthy control animals. The frequency of CD4^+^ T cells producing antigen-specific IL-2 after PPDA stimulation was 0.263%, which was 0.218% higher than that of the control group. The frequency of CD4^+^ T cells producing antigen-specific IL-2 after PPDB stimulation was 0.432%, which was 0.218% higher than in the control group. The frequency of CD4^+^ T cells producing antigen-specific IL-2 after CE stimulation was 0.405%, which was 0.452% higher than in the control group.

In addition, it can also be used to detect the secretion of IFN-γ after stimulating cells with specific stimulating agents such as avian tuberculin (PPDA), bovine tuberculin (PPDB), and early secretory antigen target 6/culture filtrate protein 10 (ESAT-6/CFP10) to diagnose bovine tuberculosis. Matteis G et al. [25] extracted bovine peripheral blood and used PPDA, PPDB, and ESAT-6/CFP10 to stimulate the blood. After that, 100 μL stimulated blood was cultured at 4 °C for 30 min, labelled anti-bovine CD4 was added, and red blood cells were lysed by adding 1 mL of ammonium chloride buffer, washed, centrifuged, resuspended, and cultured at 4 °C for 20 min. After two further cycles of washing with 800 μL Perm/Wash buffer (BD Biosciences) and centrifugation at 400× *g* for 10 min, the permeated cells were conjugated with anti-IFN-γ Alexa Fluor 647 and incubated for 30 min at 4 °C. The cells were washed, centrifuged, resuspended, and finally analysed for IFN-γ secretion by flow cytometry. By ROC analysis, the percentage of IFN-γ^+^CD4^+^ T cells in infected animals was significantly higher than that in uninfected animals. The cut-off value of the frequency of specific IFN-γ^+^CD4^+^ T cells stimulated by PPDA was 0.49%, the sensitivity was 78.6%, and the specificity was 100%; the cut-off value of the frequency of specific IFN-γ^+^CD4^+^ T cells stimulated by PPD8 was 0.32%, the sensitivity was 94.4%, and the specificity was 57.1%; the cut-off value of the frequency of specific IFN-γ^+^CD4^+^ T cells produced by ESAT-6/CFP10 was 0.48%, the sensitivity was 62.5%, and the specificity was 77.8%; the cut-off value of the frequency of specific IFN-γ^+^CD4^+^ T cells in uninfected animals was 0.26%, the sensitivity was 100%, and the specificity was 55.6%.

Studies have shown that BTB can also be diagnosed by direct immunofluorescence staining of surface antigen epitopes on peripheral blood monocytes (Figure 2). Elsayed et al. [26] used real-time PCR, traditional PCR, and flow cytometry to compare the detection rate of bovine tuberculosis. The results showed that the sensitivity and specificity of real-time PCR and flow cytometry were 100% and 100%, respectively, while the sensitivity and specificity of conventional PCR for different RDs were 81.48% and 100%, respectively. Flow-cytometry sensitivity and specificity were determined by direct binding of peripheral-blood monocyte surface epitopes to primary antibodies (monoclonal antibodies corresponding to CD4, CD8, CD8 alpha, WC1^+^ δγTCR1 chain, and CD2). Flow cytometry showed that the numbers of CD4^+^, CD8^+^, WC1^+^δγ, and CD2^+^ cell phenotypes (4900 ± 461, 4980 ± 404, 4500 ± 346, and 4390 ± 327, respectively) isolated from tuberculin-positive cattle were higher than those from tuberculin-negative cattle in terms of the number of CD4^+^, CD8^+^, WC1^+^δγ, and CD2^+^ cell phenotypes (4300 ± 230, 4550 ± 28, 4350 ± 202, and 3950 ± 548, respectively). The above studies indicate that BTB can be diagnosed by direct staining.

Bovine tubercular granuloma is the hallmark lesion of BTB caused by Mycobacterium bovine infection, and granuloma is the key to successful control of tuberculosis, which prevents the spread of pathogens through the containment of cells and fibrotic layers [57]. After invasion of the host by Mycobacterium tuberculosis, the host induces a cellular immune response to produce T cells, which are essential for the successful suppression of Mycobacterium tuberculosis by macrophages in granulomas (Figure 3). Many T cell subsets respond to a variety of Mycobacterium tuberculosis antigens [58]. With the increase in the number of bacteria, further humoral immunity is produced, and the animals infected with tuberculosis cannot be detected in time by antibody detection alone. Therefore, some flow methods based on intracellular IFN-γ and IL-2 detection have been established in the diagnosis of bovine tuberculosis, which are helpful for early detection and timely control of the disease to reduce the loss. In addition, the use of flow cytometry to detect intracellular cytokines can also determine the infection status of animals, which is helpful for disease control and treatment. 

## 4. Conclusions and Prospects

Currently, flow cytometry is widely used in immunology and virology, with good results. However, it still has many shortcomings, such as the need for large sample volumes and expensive reagents. Therefore, how to improve its detection performance is still an important issue. Attention should be paid to the following aspects: Firstly, research on experimental reagents and instruments should be strengthened. Flow cytometry can detect a variety of antigen and antibody conjugates in cells, but there are still some problems with detection reagents, such as complex preparation methods and imperfect labelling technology. Therefore, research on reagents and instruments should be intensified, and the working principles and detection methods should be continuously improved. Second, multiple trials were conducted to reduce costs. Flow cytometry has the advantages of small size, high sensitivity, and simple operation. Although the price is cheaper than NGS, the reagents and instruments required for this method are still expensive for farmers, limiting its application in agriculture and animal husbandry. Therefore, flow cytometers that are low-cost and easy to use should be developed to meet the needs of agricultural and animal husbandry production and scientific research.

In conclusion, the detection of bovine epidemics has always been a hot topic in the cattle farming industry. Flow cytometry has been widely used in the diagnosis of bovine plague due to its advantages of rapid detection speed and identification of cell survival status. Over time, flow cytometry is constantly updated and iterated, and the problems existing in the detection of flow cytometry will eventually be solved one by one.

## Figures and Tables

**Figure 1 viruses-15-01378-f001:**
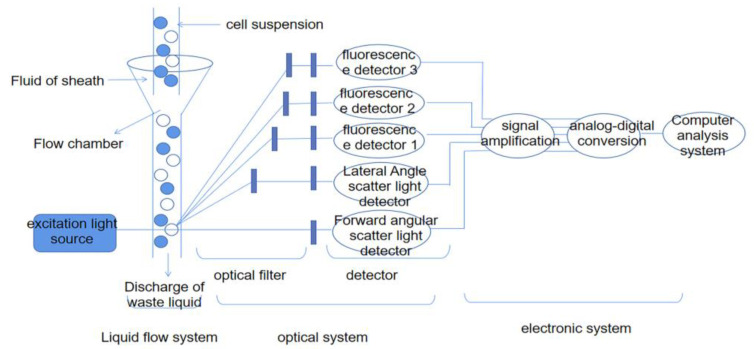
Flow cytometry structure.

**Figure 2 viruses-15-01378-f002:**
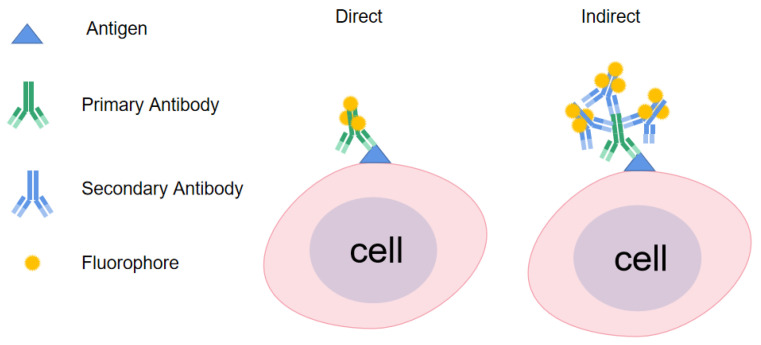
Direct dyeing and indirect dyeing.

**Figure 3 viruses-15-01378-f003:**
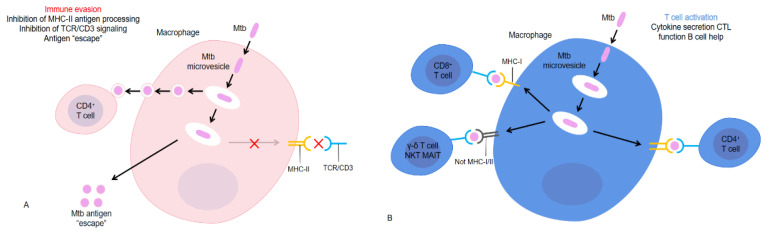
Evasion of T cell recognition versus T cell activation by Mtb-infected antigen-presenting cells. The paradox of T cell responses to Mtb is that, on the one hand, (**A**) Mtb antigens, when properly processed by activated antigen-presenting cells, elicit a broad range of T cell responses in LTBI-diseased animals. This involves many T cell subsets responding to various antigens. These MTB-activated T cells mainly secrete Th1 cytokines and chemokines, have cytotoxic T lymphocyte (CTL) functions, and help B cells. On the other hand, (**B**) Mycobacterium tuberculosis, which is sheltered by macrophages, can use several mechanisms to interfere with T cell recognition.

**Table 1 viruses-15-01378-t001:** The disease detection techniques and their methods discussed in this paper.

Disease	Test Method	Instrument	Programme	References
BVD	PrimeFlow RNA	BD LSRII flow cytometer	Detection of BVDV RNA in BVDV-infected cell lines	[15]
	PrimeFlow RNA	BD FACSymphony™ A3 flow cytometer	Detection of CD25 (IL-2 receptor) and intracellular IFN-γ in T cell subsets	[16]
	flow cytometer	CytoFLEXflow cytometer	Detection of lymphocyte counts	[17]
BLV	flow cytometer	flow cytometer	Detection of telomere length	[18]
	flow cytometer	FACSCalibur flow cytometer	Detection of BLV-gp51 expression in the blood of dairy cows	[19]
	flow cytometer	flow cytometer Navios	Assay for IL-10 and IFN-γ in MDCs	[20]
Bovine brucellosis	flow cytometer	CytoFLEX flow cytometer	Measurement of total lymphocytes, B-lymphocytes, and T-lymphocytes	[21]
	flow cytometer	flow cytometry	Detection of IFN-γ^+^ T and IFN-γ^+^ CD4^+^ T cell numbers	[22]
	flow cytometer	BD FACSymphony^TM^ A5 flow cytometer	Observation of CD4^+^ T cell proliferation and frequency of IFN-γ production	[23]
BTB	flow cytometer	FACSCalibur flow cytometer	Detection of CD4^+^ T cell numbers specific for IL-2	[24]
	flow cytometer	CytoFLEX flow cytometer	Detecting IFN-γ secretion	[25]
	flow cytometer	Becton Dickinson FACS Caliber flow Cytometer	Detection of CD4^+^, CD8^+^, WC1^+^δγ, and CD2^+^ cell phenotype numbers in cattle	[26]

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
