# Peer review of "Application of Flow Cytometry in the Diagnosis of Bovine Epidemic Disease"

_viruses, 2023, doi:10.3390/v15061378_

Round 1

Reviewer 1 Report

Dear Authors,

The manuscript show the advantages and disadvantages of flow cytometry in the detection of ruminants diseases. However we recommend to review some topics, mainly BVDV, bovine tuberculosis and verify some references that we could not find on Pubmed or other websites. Please verify the possibility to change citations.

And may explore about economic benefitials of this technique for farmers. Is it expensive, but how much? Is it more expensive in comparison with a realtime PCR or NGS?

As for English, please review a few paragraphs. In the text I highlighted BVD, bovine leukosis which was not clear.

Author Response

Dear reviewers, I have revised all the comments you have made. The first revision is located in lines 98-109 and 272-280 of the article, and the second revision is located in lines 378-381. If there are any shortcomings, I hope you can point them out, and I will keep improving them.

Reviewer 2 Report

This paper reviewed several studies that utilized flow cytometry and different assays in diagnosing bovine epidemic diseases, it outlined the recent research and offered valuable insights for future applications in this field. The review is of certain value and interest for publication.  

However, it could be improved to a higher level if the authors can dive deeper into the instrument development aspect. For example, discuss the drawbacks of the current instruments and possible directions for improvement.

To be more specific, the authors mentioned that flow cytometry needs large sample volumes and a high concentration of cells. Actually, there are already some developments in in vivo flow cytometry that resolved these issues by running the detection directly in vivo without blood drawing. For example,

a.       Microscopy-based in vivo flow cytometry which scans the blood vessels in thin tissue:

1.       J. Novak, I. Georgakoudi, X. Wei, A. Prossin, C. P. Lin, "In vivo flow cytometer

for real-time detection and quantification of circulating cells," Optics letters,

vol. 29, no. 1, pp. 77-79, 2004.

2.       Irene Georgakoudi, Nicolas Solban, John Novak, William L. Rice, Xunbin Wei,

Tayyaba Hasan, Charles P. Lin, "In vivo flow cytometry: a new method for

enumerating circulating cancer cells," Cancer Research, vol. 64, pp. 5044-5047,

2004

b.       Diffusive in vivo flow cytometry, which overcomes the limitation of thin tissue and can be applied on large limbs of animals:

 3. Vivian E. Pera, Xuefei Tan, Judith M. Runnels, Neha R. Sardesai, Charles P. Lin,

Mark Niedre, "Diffuse fluorescence fiber probe for in vivo detection of circulating

cells," Journal of Biomedical Optics, vol. 22, no. 3, p. 037004, 2017.

4. Xuefei Tan, Roshani Patil, Peter Bartosik, Judith M. Runnels, Charles P. Lin, Mark

Niedre, "In Vivo Flow Cytometry of Extremely Rare Circulating Cells," Scientific

Reports, vol. 9, no. 1, p. 3366, 2019

In vivo flow cytometry avoids the need for blood drawing, and some of them have high sensitivity that can detect cells at very low concentrations (Ref. 4) and hence can be used for early detection of disease. Current in vivo flow cytometry methods, although have their own limitations, they can possibly bypass the drawbacks of in vitro methods. Besides what was discussed above, in vitro methods have a high possibility to miss the early stage of infection, blood drawing may cause disturbance to animals, blood sample processing may cause loss and damage to cells.

Other minor corrections to this article:

1.       Line 13: “the configuration and working principle of flow cytometry were introduced”

2.       A table to list and compare the studies and their methods discussed in this article would be very helpful, including the disease/virus, the instrument (flow cytometry also has many versions), protocols, etc.

3.        Line 326, the authors mentioned “some flow methods based on intracellular IFN-γ and IL-2 detection have been established in the diagnosis of bovine tuberculosis” but didn’t reference any research. 

The English can be improved but doesn't cost too much effort to read. 

Author Response

Dear reviewers, I have added the section you suggested about in vivo flow cytometry in lines 62-68 of the article. Regarding your minor corrections, the first comment is located in lines 13-15 of the article and the second icon has been added to line 90 of the article. For your third comment, which is my comment on the bovine TB assay, the information about IL-2 is located in the literature mentioned in lines 295-309 of the article. the information about IFN-γ is located in the literature in lines 310-329 of the article. I would like to thank you for your valuable suggestions. If there are still deficiencies, please point them out. I will continue to correct them.

Round 2

Reviewer 1 Report

Dear Authors,

Thank you to reviewed our suggestions and accepted some of them.

In Line 81, on Figure 1, the image not appear. Please check again.

Author Response

Dear Reviewer.
Hello, I have re-added Figure 1 in line 81 of the article. I would like to thank you from the bottom of my heart for your suggestions on my article and hope that you will continue to criticise me if there are any shortcomings.

Reviewer 2 Report

The manuscript looks better now, except some new issues occurred:

1.       Line 49, tile of section 2 missing bracket for “FCM”.

2.       Figure 1 is lost.

3.       It would be better to label the reference in Table 1.

4.       There are some instructions for each section at the bottom of the script, for example, “Supplementary Materials”,   “Author Contributions”, etc. Please delete the instructions or the whole section if you don’t need them. 

Please do not use languages other than English for communication purposes, as the reviewer(s) may have difficulty understanding them. Thanks!

Author Response

Dear Reviewer.
Hello, I have re-added bracket for "FCM" in line 49 of the article and also re-added Figure 1, and I have also added references to the table. I would like to thank you from the bottom of my heart for your suggestions on my article, and I hope you will continue to criticize me if there are still any shortcomings.